# Bipolar Spectrum Symptoms in Patients with Fibromyalgia: A Dimensional Psychometric Evaluation of 120 Patients

**DOI:** 10.3390/ijerph192416395

**Published:** 2022-12-07

**Authors:** Vittorio Schweiger, Giovanni Perini, Lidia Del Piccolo, Cinzia Perlini, Valeria Donisi, Leonardo Gottin, Alvise Martini, Katia Donadello, Giovanna Del Balzo, Valentina Moro, Erica Secchettin, Enrico Polati

**Affiliations:** 1Department of Surgery, Dentistry, Paediatrics and Gynaecology, Pain Therapy Center, University of Verona, 37124 Verona, Italy; 2Department of Neurosciences, Biomedicine and Movement Sciences, Clinical Psychology, University of Verona, 37124 Verona, Italy; 3Department of Medicine and Public Health, University of Verona, 37124 Verona, Italy; 4NPSY-Lab.VR, Department of Human Sciences, University of Verona, 37124 Verona, Italy

**Keywords:** fibromyalgia, chronic pain, bipolar spectrum, depressive symptoms, manic symptoms

## Abstract

Background: Fibromyalgia Syndrome (FMS) is characterized by chronic widespread pain, fatigue, unrefreshing sleep and cognitive dysfunction. Depressive and manic symptoms are often reported in FMS patients’ history. The aim of this study was to evaluate the prevalence of bipolar spectrum symptoms (BSS) and to correlate these with quality of life (QoL) scores and antidepressant treatment. Methods: From October 2017 to July 2018, a battery of QoL questionnaires (FIQ, PSQI and SF-12) was administered to 120 FMS patients after a clinical examination. The MOODS-SR lifetime questionnaire was then remotely administered to the patients included in the study. Results: The presence of depressive and manic lifetime symptoms was found, in line with the results of the available literature. A correlation was found between the history of depressive symptoms and the severity of FIQ and SF-12 scores. Despite a low statistical strength, a trend toward a correlation between a history of manic symptoms and SNRI treatment was detected. Conclusions: The correlation between the MOOD—depressive domains and poor QoL is in line with the available literature. Further studies are needed to corroborate these findings and to elucidate the relationship between manic symptoms and SNRI treatment.

## 1. Introduction

Fibromyalgia syndrome (FMS) is a multifaceted disorder characterized by chronic widespread pain, fatigue, unrefreshing sleep and cognitive dysfunction [1]. The prevalence of FMS is around 2–6% of the general population according to different epidemiological reports [2]. The female/male ratio is reported to be 5:1, with a typical onset in the 30–60 year interval of age [3]. Psychiatric disturbances in patients with FMS were thoroughly investigated. While depressive disorders are commonly known to be associated with FMS, manic disturbances were investigated only recently [4]. According to the DSM5, the term bipolar disorder (BD) defines a group of psychiatric disorders characterized by extreme fluctuations in mood, energy, cognition and general functioning [5]. The incidence of BD in FMS was reported in 0–70% of patients, depending on the diagnostic criteria [6,7]. Some authors, however, underlined that the DSM5 diagnostic criteria were too restrictive, leading to exclusion from the BD diagnosis of patients with sub-threshold symptoms [8]. For this reason, the definition of bipolar spectrum symptoms (BSS) was introduced, referring to a dimensional “continuum” between depression and mania without the categorical distinction described in the DSM5 [9]. To assess the BSS, the SCI-MOODS and its self-reporting version, the MOODS-SR questionnaire, were developed [10,11]. At present, MOODS-SR is the only self-reporting instrument exploring, alongside depressive symptoms, the manic/hypomanic dimensions and their associated features. From a therapeutic perspective, the presence of BSS in the history of FMS patients is of pivotal importance for the possibility that drugs such as duloxetine, a first-line pharmacological treatment for this syndrome, can cause or exacerbate manic switching [12,13]. In this prospective observational study, we evaluated the presence of BSS in a population of FMS patients. Moreover, we correlated BSS with the usual QoL tools used in FMS and with the antidepressant treatment at the time of observation.

## 2. Materials and Methods

### 2.1. Participants

The patients in the study population were referred to the Pain Therapy Centre of the Verona University Hospital with a diagnosis of FMS according to the 2016 ACR (American College of Rheumatology) Revised Diagnostic Criteria (WPI—widespread pain index ≥7 and SS—symptom severity ≥5 or WPI 4–6 and SS ≥9; pain in at least 4 of 5 corporeal regions, excluding jaw, chest and abdominal pain; symptoms for at least 3 months) [14]. The eligible population included new and already followed patients who came to the centre for clinical examination.

### 2.2. Questionnaires

Before the clinical evaluation, the FIQ, PSQI and SF-12 questionnaires were submitted to all enrolled patients. The Fibromyalgia Impact Questionnaire (FIQ) is the most used instrument for evaluating QoL in FMS patients, which has also been validated in the Italian population [15]. The questionnaire consists of 21 individual questions based on an 11-point numeric rating scale from 0 to 10, with 10 being “worst”. All questions, related to the past 7 days, are divided into three domains (function, overall impact and symptoms), with a maximum total score of 100. The higher the total score, the more severe the dysfunction for each single patient. The Pittsburgh Sleep Quality Index (PSQI) was introduced to evaluate the sleep quality of patients considering the duration of sleep, sleep latency, and the frequency and severity of some sleep-correlated problems [16]. It consists of 19 questions, each scoring from 0 to 3, with a maximum final score of 21, corresponding to the worst sleep quality. This questionnaire was validated in the Italian population [17]. The short-form 12 (SF-12) was introduced in 1996 with the purpose of creating a synthetic version of the SF-36 and is used to evaluate the perceptions of the patient about her/his psychological and physical condition [18]. Through use of an algorithm, two indexes can be calculated, namely the PCS-12 (Physical Component Summary) and the MCS-12 (Mental Component Summary). If the PCS-12 score is high, this identifies individuals without physical limitations, disability or a decrease in general well-being, while a low score indicates strong limitations in physical or social activities, pain or frequent fatigue. If the MCS-12 score is high, this describes a positive psychological attitude and the absence of psychological distress; on the contrary, low scores indicates severe psychological distress and social and personal disability. This questionnaire has been validated in the Italian population within the IQOLA Project [19]. After signing the informed consent form, each patient was asked to fill in the MOODS-SR lifetime questionnaire, which was sent via e-mail by Cloud Survey Company, which notified us about the request to participate in the study on the company’s website. The Mood Spectrum Disorder—Self Report questionnaire (MOODS-SR) was derived from the structured clinical interview version (SCI-MOODS) to assess both the overt and subtle components of depression and mania along a continuum of various psychopathological dimensions and different levels of mood dysregulations. The MOODS instruments focus on the presence of manic and depressive symptoms, and the traits and lifestyles that characterize this dysregulation, and investigate both fully syndromic and subthreshold mood disturbances [11]. The questionnaire is composed of 161 items, with the answers being “No” (score 0) and “Yes” (score 1), indicating absence or presence of one or more periods of specific situations (lasting at least of 3–5 days duration). The MOOD questionnaire is divided into three main sections (manic, depressive and rhythmicity); the first two are divided into three subsections (mood, energy and cognition). The seven impairment items are not scored, and items with multiple responses are coded as “1” if at least one of the responses is endorsed positively and “0” if it is not. The higher the number of positive responses, the more severe the dysfunction for each patient. While a threshold for the total score or for any of the subsections has not been determined, being a dimensional scale, the mean scores for MOOD—depressive, MOOD—manic and MOOD—rhythmicity in healthy controls were specified in the validation study [11].

### 2.3. Data Collection

We collected demographic and clinical data, including age, gender, educational years, employment, marital status and ongoing antidepressant drug treatment. The mean daily pain intensity was evaluated with the VAS score (visual analogue scale, 0–100) reported by patients during the medical evaluation and in the last 7 days extracted from the FIQ questionnaire. The scores of the four questionnaires were also collected.

### 2.4. Statistical Analysis

Statistical analysis was conducted using RStudio, Version 1.3.959 (250 Northern Ave, Boston, MA, USA). Descriptive statistics were reported in terms of the mean and standard deviation for quantitative variables, and in terms of absolute frequencies and percentages for qualitative variables. The scores and distributions of the scales were reported in terms of the median and interquartile range. All tests were two-sided, with the significance level set at 5%. Each questionnaire was considered as an independent variable. The MOODS-SR results were collected in absolute numbers, as indicated by the scoring algorithm of the questionnaire. Percentages were obtained by dividing the number of positive answers for each domain by the maximum score for each domain (excluding the items not scored, namely Items 28, 57, 67, 80, 108, 131 and 161) and were then correlated with the results of the QoL tests (FIQ, PSQI and SF-12) using Spearman’s correlation coefficient analysis. Among the patients who responded to the MOODS-SR lifetime questionnaires, the patients taking antidepressant drugs (TCAs, SNRIs or SSRIs) were selected and the data were correlated with the MOODS scores. The statistical significance was assessed with *t*-tests for paired data.

### 2.5. Ethics and Consent

All participants were informed about the details of the research and signed the informed consent form before the enrolment. The study was approved by the local clinical research committee (RED protocol, ID 1751CESC).

## 3. Results

From October 2017 to July 2018, 120 patients (113 female, 7 male) were enrolled. The evaluation of the returned online survey showed that 22 patients did not complete the MOODS-SR lifetime questionnaire and were not included in the analysis. The demographic characteristics of the whole population and dropouts are reported in Table 1. Dropouts turned out to be slightly more educated and were more likely to be married than the study sample.

The mean VAS score (visual analogue scale, 0–100) reported by the patients during the medical evaluation was 72.7 ± 25, slightly lower than value extracted by the FIQ questionnaire completed during the same evaluation, although this difference was not statistically significant. Antidepressant treatment at the time of evaluation was highlighted in 40 patients (40.8%), with a prevalent use of SSRIs (17 patients, 42.5%), followed by TCAs (13 patients, 32.5%) and SNRIs (10 patients, 25%). The median scores of the submitted questionnaires among the analysed population (98 patients) are reported in Table 2.

An analysis of the correlations between the MOODS domains and the scores of the QoL questionnaires showed that the ratio between positive answers and total depressive domain questions (MOOD-D) was significantly correlated with the FIQ and SF-12 scores, while the total manic domain questions (MOOD-M) were not significantly correlated with severity of all the QoL scores considered. The significant correlations are shown in Figure 1 and Figure 2.

Moreover, the correlation analysis between antidepressant treatment and the absolute values of the MOOD-SR showed that the mean MOOD—manic scores (MOOD-M) were higher in patients treated with SNRIs compared with TCAs and SSRIs, though this difference was not statistically significant (*p* = 0.2) (Figure 3).

## 4. Discussion

This observation conducted by the online administration of the MOOD-SR questionnaire in a cohort of FMS patients showed the presence of both subthreshold depressive and manic lifetime symptoms in this population. The values of the absolute mean MOOD-D and MOOD-M domains in our sample (20.0 and 17.0 respectively) were almost exactly in line with the results of similar studies conducted in Italy on analogous FMS samples (MOOD-D, 21.6; MOOD-M, 16.5 in 167 patients; MOOD-D, 23.7; MOOD-M, 16.9 in 40 patients) [20,21]. In our study, conducted on 98 patients, a significant correlation between the total lifetime depressive domain of the MOOD-SR (MOOD-D) was related with the current severity of FIQ and SF-12 scores. This is in line with the first previous prospective, observational study on a similar sample (*n* = 167), which showed a positive correlation between the number of lifetime depressive MOOD scores and higher severity of pain and worse QoL [20]. In the same study, Dell’Osso and colleagues also found that lifetime manic symptoms were statistically related with current pain severity measured by the VAS score extracted by FIQ, with the current FIQ total scores and with current poor physical and mental scores in the SF-36, both in the whole sample and also in patients without bipolar disorder diagnosed according to the DSM-IV-TR criteria. In a more recent study though (*n* = 40), the same authors did not confirm this finding. In fact, no statistically significant correlation between the MOOD-M components and the bodily pain SF-36 subscale was found [21]. It is worth mentioning that both previous Italian FMS samples included only females, while our sample, although M/F unbalanced, was not gender-specific. The correlation between the MOOD-SR domains and sleep quality measured by the PSQI did not show statistical significance in our sample. In our patients undergoing antidepressant treatment, the mean MOOD manic scores (MOOD-M) appeared to be higher in patients treated with SNRIs compared with TCAs and SSRIs. While this evidence was not strongly statistically supported (*p* = 0.2), the finding of slightly higher manic subthreshold symptoms among FMS patients who take SNRIs rather than SSRIs or TCAs is of particular interest. The data should be interpreted with caution, as the administered MOODS questionnaire was for lifetime questions, whereas the AD treatment took place only at the time of the interview. For this reason, it is impossible to measure how much of this trend toward hyperactivity and disinhibition was caused by the use of stimulating drugs. Nevertheless, the specificity of higher scores in the manic domain for the AD class most at risk of manic switches (SNRIs > SSRIs) seems to suggest an iatrogenic hypothesis [22].

The limitations of our study are the lack of a control group and the different time-span covered by the psychometric questionnaire (MOODS-SR lifetime) and the pain/QoL questionnaires, leading to some uncertainty about the possible temporal and causal correlations between the results. The small size of the sample and the M/F disproportion are other possible confounding factors that could have led us to overestimate the reported correlations. Moreover, a relevant dropout from the MOOD-SR survey was registered. This was probably due to several factors, such as the test’s length, the emotional engagement related to some questions, the email modality of the test and the explicit research goal with no direct consequences for the patients’ QoL. However, the administration of a fine-grained tool for the investigation of mood sensitivity and behavioural patterns in the population of FMS patients, which has usually only been superficially described in terms of their behavioural and psychological characteristics, remain of particular interest for pain clinicians, particularly for the frequent prescription of high-potency noradrenergic compounds in this population. From the patients’ perspective, the routinary use of such tests in paper form as a part of the clinical assessment, with the explicit aim of better drug prescription (e.g., no SNRIs in BSS) may improve the acceptance of these investigations in the future and increase the attraction of these assessments in the clinical setting.

## 5. Conclusions

Our study on FMS patients showed a correlation between a history of depressive symptoms measured with the MOOD-SR questionnaire and poor QoL. These observations are in line with the available literature, but further studies on larger populations to corroborate these findings are needed. The administration of the “last month” or even the “last week” versions of the MOODS-SR could better investigate the possible direct effects of these behavioural patterns on pain and QoL in the FMS patient population and the possible iatrogenic (SNRIs) induction of manic symptoms in some of these patients.

## Figures and Tables

**Figure 1 ijerph-19-16395-f001:**
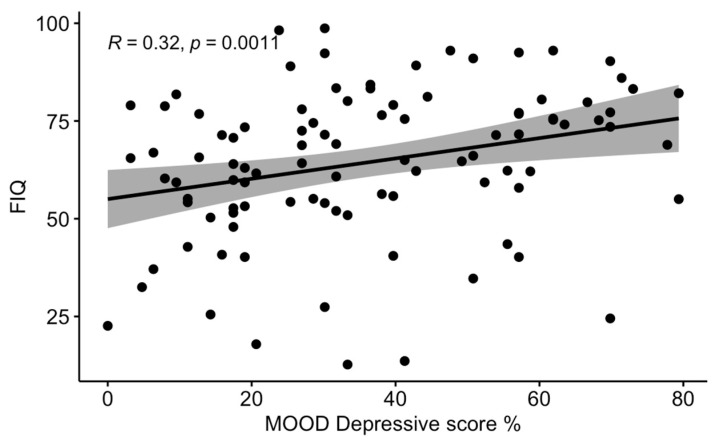
Correlation between the MOOD—depressive domain (percentage values) and the FIQ score.

**Figure 2 ijerph-19-16395-f002:**
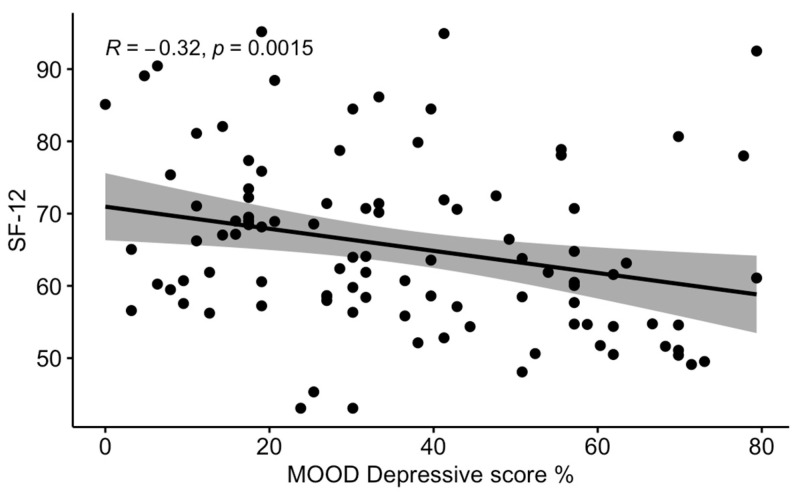
Correlation between the MOOD—depressive domain (percentage values) and the SF-12 score.

**Figure 3 ijerph-19-16395-f003:**
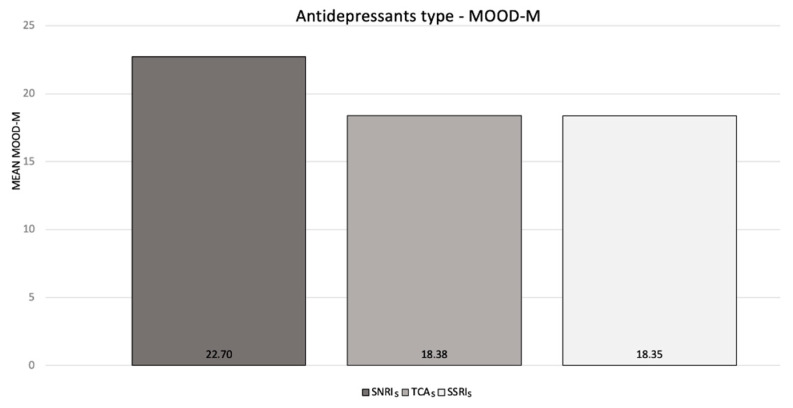
Antidepressant treatments (SNRIs, TCAs and SSRIs) and median MOOD-SR maniac component values. SNRIs, serotonin noradrenaline reuptake inhibitors; TCAs, tricyclic antidepressants; SSRIs, selective serotonin reuptake inhibitors.

**Table 1 ijerph-19-16395-t001:** Demographic characteristics of the whole population (*n* = 120).

	Sample (*n* = 98)	Dropouts (*n* = 22)	*p*
Age (mean ± SD)	51.2 ± 10.4	50.8 ± 11.5	ns
M:F ratio	1:16	1:18	ns
Education (years ± SD)	12.41 ± 3.58	15.27 ± 4.79	0.02
Marital status (Yes)	58.2%	36.4%	0.06
Employment (Yes)	60.2%	50%	ns

ns = not statistically significant.

**Table 2 ijerph-19-16395-t002:** Median values (25–75%) of the questionnaires administered to the analysed population (*n* = 98).

	(*n* = 98) Median (IQR)
FIQ (0–100)	65.90 (54.23–77.80)
VAS-FIQ (0–100)	77.50 (60.00–95.00)
MOOD—depressive (M-E-C)	20.00 (11.25–34.75)
MOOD—manic (M-E-C)	17.00 (12.00–24.00)
MOOD—rhythmicity	14.00 (10.00–17.00)
PSQI	13.00 (10.00–16.00)
SF-12	63.35 (57.15–71.78)
PCS 12	29.45 (25.17–33.00)
MCS 12	35.34 (30.01–40.38)

FIQ, Fibromyalgia Impact Questionnaire; VAS, visual analogue scale; M-E-C, mood–energy–cognition subsections; PSQI, Pittsburgh Sleep Quality Index; SF-12, Short Form 12; PCS, Physical Component Summary; MCS, Mental Component Summary.

## Data Availability

Not applicable.

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
