# Peer review of "Bipolar Spectrum Symptoms in Patients with Fibromyalgia: A Dimensional Psychometric Evaluation of 120 Patients"

_ijerph, 2022, doi:10.3390/ijerph192416395_

Round 1

Reviewer 1 Report

Dear Author

This study is of great clinical relevance, as it allows us to evaluate the association between lifetime bipolar symptoms, quality of life and antidepressant treatment for FMS. The manuscript is also well written.

My suggestions for this manuscript are set out below.

1.         There seems to be a dissociation between the conclusions of the abstract and the conclusions of the manuscript. Please try to make them as consistent as possible.

2.         MOOD-D, MOOD-M and Rhythmicity results are listed in Table 2, but a detailed description of what each represents is needed.

3.         Figure 1 and 2 should be named on the horizontal axis. The font should also be larger.

4.         The line should be broken at the limit of the discussion (P6;L235).

Author Response

  1. There seems to be a dissociation between the conclusions of the abstract and the conclusions of the manuscript. Please try to make them as consistent as possible.

The abstract was modified according conclusion’s section

  1. MOOD-D, MOOD-M and Rhythmicity results are listed in Table 2, but a detailed description of what each represents is needed.

Details about Table 2 were added; moreover, the method’s section was implemented

  1. Figure 1 and 2 should be named on the horizontal axis. The font should also be larger.

Figures were modified according the suggestions of the reviewer

  1. The line should be broken at the limit of the discussion (P6;L235).

Thanks, it is add.

Reviewer 2 Report

I reviewed the paper entitled " Bipolar spectrum symptoms in patients with Fibromyalgia: a dimensional psychometric evaluation on 120 patients".

There are some major remarks which should be addressed:

1- The aim of the study is not obviously declared, and the title does not match the aims and results of the study. 

2- It is very hard to follow the aims and findings of the study. 

3- Statistical analysis is too superficial and regression models considering baseline adjustments could be measured. 

4- Only Mood-Depressive is assessed and correlations of Mood-manic are missed.

5- Why did the authors use the mentioned questionnaires? This selection should be scientifically and methodologically stated. 

6- A control group can help better compare and prove the results. 

7- The conclusions are not methodologically supported by the results. 

Author Response

  1. The aim of the study is not obviously declared, and the title does not match the aims and results of the study. 

The abstract was updated. The aim of the study was better clarified with more details.

  1. It is very hard to follow the aims and findings of the study. 

The aims and the findings of the study were more detailed to improve clarity

  1. Statistical analysis is too superficial and regression models considering baseline adjustments could be measured. 

There are no obvious corrective factors to perform the suggested regression analysis. Moreover, the study purpose was to evaluate this population in the real clinical scenario

  1. Only Mood-Depressive is assessed and correlations of Mood-manic are missed.

   Correlations between MOOD manic domain were missed because not statistically significant

  1. Why did the authors use the mentioned questionnaires? This selection should be scientifically and methodologically stated.

Details about the use of the mentioned questionnaires are reported in the methodological section. The reported questionnaires (FIQ-R, PSQI and SF12) are the most used questionnaires in the FMS population worldwide, also for research purposes

  1. A control group can help better compare and prove the results. 

Thanks for advice. This point is added as further limitation of the study.

  1. The conclusions are not methodologically supported by the results. 

The conclusions have been implemented, thanks for suggestions

Round 2

Reviewer 2 Report

My comments have been addressed.